# Rapid Non-Destructive Analysis of Food Nutrient Content Using Swin-Nutrition

**DOI:** 10.3390/foods11213429

**Published:** 2022-10-29

**Authors:** Wenjing Shao, Sujuan Hou, Weikuan Jia, Yuanjie Zheng

**Affiliations:** School of Information Science and Engineering, Shandong Normal University, Jinan 250358, China

**Keywords:** non-destructive detection technique, food nutrition, machine vision, deep learning, nutrition evaluation

## Abstract

Food non-destructive detection technology (NDDT) is a powerful impetus to the development of food safety and quality. One of the essential tasks of food quality regulation is the non-destructive detection of the food’s nutrient content. However, existing food nutrient NDDT performs poorly in terms of efficiency and accuracy, which hinders their widespread application in daily meals. Therefore, this paper proposed an end-to-end food nutrition non-destructive detection method, named Swin-Nutrition, which combined deep learning and NDDT to evaluate the nutrient content of food. The method aimed to fully capture the feature information from the food images and thus accurately estimate the nutrient content. Swin-Nutrition resorted to Swin Transformer, the feature fusion module (FFM), and the nutrient prediction module to evaluate nutrient content. In particular, Swin Transformer acted as the backbone network for feature extraction of food images, and FFM was used to obtain the discriminative feature representation to improve the accuracy of prediction. The experimental results on the Nutrition5k dataset demonstrated the effectiveness and efficiency of our proposed method. Specifically, the mean value of the percentage mean absolute error (PMAE) for calories, mass, fat, carbohydrate, and protein were only 15.3%, 12.5%, 22.1%, 20.8%, and 15.4%, respectively. We hope that our simple and effective method will provide a solid foundation for the research of food NDDT.

## 1. Introduction

The safety and health of food has become a constant public concern, and the food nutrient content is an essential indicator of food health [1,2]. Understanding and evaluating the content of various nutrients [3,4] is important for public self-health management and disease prevention. Traditional nutrition detection requires professionals to detect nutritional components of food. Due to its complicated operation process, it is infeasible for the public to accurately know the nutritional information in daily diet. At present, food NDDT [5,6,7] has attracted more attention because it has the advantages of not destroying the food structure when measuring food composition. Therefore, it is significant to explore a method that can both non-destructively detect the nutritional content of food and quickly understand the nutritional information to promote public dietary health.

Some traditional food nutrition detection [8,9,10] is performed by professionals using physical or chemical methods to measure calorie, carbohydrate, protein, and fat content. However, professionals were susceptible to external factors such as the environment, resulting in a lack of accuracy. In addition, the number of professionals did not meet the public demand for dietary nutrition estimation. In recent years, many researchers have used the NDDT to evaluate various food components, which has alleviated the shortage of professionals to some extent and accelerated the development of food safety and quality [11,12]. Xu et al. [13] proposed a possibilistic fuzzy c-means algorithm to accurately identify apple varieties on the assembly line. Liu et al. [14] designed a non-destructive detection device for apple sugar content based on NIR detection technology, established a prediction model for apple sugar content, and realized rapid non-destructive detection of apple sugar content. Liu et al. [15] used multi-spectral imaging combined with machine learning to perform rapid and non-destructive detection of the level and content of zinc contamination in maize, and the study has important implications for the safety of maize seeds. Gong et al. [16] collected terahertz spectra and images of intact ginkgo seeds and their slices with different water contents and used the sensitivity of terahertz waves for liquid water to study the non-destructive detection method of water content variation in ginkgo fruit. Xue and Tan [17] utilized front-face synchronous fluorescence spectroscopy to analyze the composition of flour blends rapidly and non-destructively, and used least squares regression to develop predictive models for binary and quaternary flour blends. With the development of artificial intelligence, the combination of NDDT and machine learning or deep learning methods [18,19] has made the assessment and analysis of the food’s nutritional content more efficient and accurate.

In the past few years, computer vision-based methods [11,20,21] for non-destructive detection of food nutrition have gradually emerged. Some nutrition estimation applications [22] were widely used by subscribers. The user took a picture of each meal and uploaded a picture of the food and recorded the weight of each ingredient. The applications estimated roughly the nutritional content of each meal. However, these applications [23] were only able to identify single food item and required users to manually record weight or ingredient information. This process was tedious, inaccurate, and inefficient [24]. With the development of deep learning technology, more efficient and precise methods [25,26,27] for food nutrition evaluation were proposed. Myers et al. [25] proposed a restaurant food recognition and calorie estimation system for fast and non-destructive detection of calorie content in meals. Ege and Yanai [26] used convolutional neural networks (CNN) to predict the composition and calories of food from images. Jaswanthi et al. [28] proposed a hybrid network of generative adversarial network and CNN for food segmentation, recognition, and calorie estimation, but the method only predicted a single ingredient and calculated the calorie content indirectly through volume estimation. Thames et al. [21] disclosed the *Nutrition5k* dataset and combined it with convolutional neural networks for calorie, mass, carbohydrate, fat, and protein content prediction. This method facilitated the research of nutrition evaluation based on computer vision, but failed to provide a more efficient non-destructive detection method for food nutrition. These methods simply predict the nutritional content of food or indirectly estimate the calorie content of ingredients. In addition, they fail to sufficiently take into account the specificities of the food nutrition prediction task, such as the difficulty of predicting fine ingredients, the diversity of ingredients, etc.

Considering the problems of existing methods and the specificity of food nutrition detection, we propose a Swin-Nutrition method to detect the nutritional content of food rapidly and non-destructively by analyzing the content of calories and macronutrients (fat, carbohydrate, and protein) from food images. Swin-Nutrition captures an effective feature representation for food nutrition non-destructive detection and enhances the performance of model nutrition prediction. The method adopts Swin Transformer [29] as the backbone network and combines the pyramid structure by designing the feature fusion module to fuse multi-scale feature information effectively. We exploit the long-range dependency of Swin Transformer to capture the global feature representation and introduce the feature fusion module to fuse feature maps of different resolutions. Therefore, our proposed rapid non-destructive detection model Swin-Nutrition obtained excellent performance on the *Nutrition5k* dataset [21].

## 2. Materials and Methods

### 2.1. Data Preparation

We used the RGB group data from the *Nutrition5k* dataset to explore food nutrition NDT. *Nutrition5k* dataset is captured by the *Intel RealSense D435* camera. The camera consists of four main sensors, including two infrared sensors, an infrared laser emitter, and a color sensor. The wavelength of the infrared laser emitter is 850 nm ± 10 nm. The first two are responsible for forming the depth images, and the latter takes care of forming the RGB images. The depth and RGB images eventually make up the *Nutrition5k* dataset. Some illustrative images from the RGB group and depth group from the Nutrition5k dataset are shown in Figure 1. The dataset contains 5000 different dishes with more than 250 ingredient categories. The construction of this dataset follows the principle of incrementality, where each dish is gradually added from a single ingredient to multiple ingredients, up to 35 ingredients. For the categories of ingredients and the number of pictures, *Nutrition5k* is the largest dataset in the field of food nutrition evaluation. We carefully examine this dataset and find that some of the images have serious annotation errors or food items that are not present in the images. Therefore, we clean this dataset and use the cleaned RGB group data in our experiments. The training set contains 207,576 images and the test set contains 34,934 images, with a ratio of 6:1 between the training and test sets.

### 2.2. Methods

In this section, we first describe the overall structure of our method in Section 2.2.1. Then, we introduce the feature extraction module, Swin-Nutrition’s Backbone Network, in Section 2.2.2. In Section 2.2.3, we dissect the feature fusion module, which enhances the feature representation. Finally, we introduce the multi-task loss function in Section 2.2.4.

#### 2.2.1. Overall Architecture of the Approach

The overall architecture of our network is shown in Figure 2. The network contains three components: backbone network, FFM, and nutrition predicted module. The input image is first sent to the backbone network to extract feature information. The backbone network is Swin Transformer, which obtains global and local feature representation from the images. The Swin transformer exploits stacked transformer blocks to obtain multi-level feature maps. The multi-level feature maps are fed into the FFM, which is built according to the structure of the feature pyramid. It is used to fuse multi-level feature maps and finally output a high-resolution, strongly semantic feature map. After the nutrition prediction module (global average pooling layers and FC layers), the fused feature map outputs predictions for calories, mass, and macronutrients. Swin-Nutrition not only fully extracts the feature information from RGB images for nutritional prediction but also fuses the feature maps of different layers to improve the prediction accuracy.

#### 2.2.2. Swin-Nutrition’s Backbone Network

Swin-Nutrition’s backbone network follows the structure in Swin Transformer, building four transformer blocks. Swin Transformer combines the advantages of both CNN and transformer to obtain multi-scale feature representation by stacking transformer blocks. Therefore, we use Swin Transformer as our backbone network to extract image feature information. The input image is first divided into non-overlapping patches by patch partition. The size of each patch is 4 × 4. In stage 1, the channel number of the patch is first changed to C by the linear embedding layer and then input to the Swin Transformer block. In stage 2, the model is first passed through the patch merging layer, which has the effect of downsampling the feature maps, and then are fed to the Swin Transformer block. Swin Transformer block is composed of two consecutive transformer blocks, and each transformer block consists of two layers of multi-layer perceptron (MLP), LayerNorm layer (LN), multi-headed self-attention layer, and residual connections [30]. Multi-headed self-attention is replaced by window-based self-attention (W-MSA) and shifted window-based self-attention (SW-MSA), as shown in Figure 3. W-MSA restricts the calculation of attention to each window, while the traditional transformer calculates attention based on global information. This module is proposed to reduce the computational complexity for intensive prediction tasks. SW-MSA is designed to enhance the information interaction between individual windows. Both stage 3 and stage 4 have the same modules as stage 2. The Swin Transformer utilizes a hierarchical structure where each layer reduces the resolution of the feature map and enlarges its receptive field.

After extracting image feature information from Swin-Nutrition’s backbone network, we utilize the extracted feature information for nutritional evaluation. Although Swin Transformer has powerful performance in local and global feature extraction, there are smaller ingredients in the food nutrition estimation task that are difficult to detect with the naked eye. Considering the particularity of the nutrition assessment task, we need to mine the local detailed information from images to solve the difficult problem of detecting small ingredients. Therefore, we introduce FFM to obtain multi-scale feature information and enhance the local feature representation.

#### 2.2.3. Feature Fusion Module

To improve the extraction capability of local feature of the Swin-Nutrition model, we propose the feature fusion module, as shown in Figure 2. The sizes of the feature maps output by each block of the Swin Transformer are $\frac{H}{4}\times \frac{W}{4}$, $\frac{H}{8}\times \frac{W}{8}$, $\frac{H}{16}\times \frac{W}{16}$, and $\frac{H}{32}\times \frac{W}{32}$, respectively. *H* and *W* respectively represent the height and width of the input image. We fuse the output of each layer by top-down and lateral connections, as shown in the red rectangular box in Figure 2. The top-down means that the output feature map of this stage is scaled up to the same size as the feature map of the previous stage by upsampling. Lateral connection fuses the results of upsampling with the feature maps of the same resolution generated from the top-down. We use 1 × 1 convolution to vary the number of channels in the output. In this way, the two feature maps are the same in space size. We fuse the feature maps of the higher layers with the feature maps of the adjacent layers. By this means, our model utilizes the semantic feature of the higher layers and the high-resolution information of the lower layers. The fused feature maps have richer semantic and detailed information to enhance the accuracy of nutrient prediction. Finally, the feature maps of the four layers are subjected to global average pooling to obtain the final feature maps.

#### 2.2.4. Multi-Task Loss Function

By observing the loss values for the five subtasks of calories, mass, fat, carbohydrate, and protein, we find that the loss values for mass and calories are relatively large, and the loss values of these two tasks are in different orders of magnitude from the other three subtasks. To balance the loss values of the five subtasks, we adopt the normalization method, as shown in Equation (Equation 2). In addition, our loss function uses an uncertainty weighting method [31], which is weighted according to the uncertainty of the task. Our loss function is defined as follows: (1)$$\begin{array}{cc} L_{T} & =\sum_{\tau \in T}\frac{1}{2c_{\tau }^{2}}L_{\tau }\left(x,y_{\tau },y_{\tau }^{\prime },\omega_{\tau }\right)+\ln\left(1+c_{\tau }^{2}\right) \end{array}$$(2)$$\begin{array}{cc} L_{\tau } & =\frac{\sum_{i=1}^{N}\left|{y^{\prime }}_{\tau_{i}}-y_{\tau_{i}}\right|}{\sum_{i=1}^{N}y_{\tau_{i}}} \end{array}$$
where T∈{cal,mass,protein,carb,fat}, the multitask loss is the weighted sum of the five subtask losses: the calorie regression loss $l_{\mathrm{cal}}$, the mass regression loss $l_{\mathrm{mass}}$ and the regression losses $l_{\mathrm{carb}}$, $l_{\mathrm{fat}}$, and $l_{\mathrm{protein}}$ for the three macronutrients. The loss for each subtask is used as the regression loss using the ratio of mean absolution error (MAE) to the average of all groundtruth values for that task. We weight the loss function for each subtask, where $c_{\tau }$ is the learnable parameter and $\ln\left(1+c_{\tau }^{2}\right)$ is the regularization term to avoid generating negative loss values. $y_{\tau_{i}}^{\prime }$, $y_{\tau_{i}}$ denote the predicted and groundtruth values of each subtask, respectively. We apply an uncertainty-weighted loss function to the five subtasks, and the task is relatively stable when the uncertainty of the subtask is small. Therefore, we should assign a larger weight to the task. On the contrary, for tasks with higher uncertainty, the more noise in the task-related output, the harder the task is relatively to learn. Thus, we should weaken the weight of that subtask. In this way, we can improve the efficiency of multi-task learning and the accuracy of prediction.

### 2.3. Evaluation Metrics

We measure the accuracy of the regression of calories, mass, and macronutrients using the percentage mean absolute error (PMAE) [21], which is expressed as follows: (3)$$\begin{array}{cc} \mathrm{MAE} & =\frac{1}{N}\sum_{i=1}^{N}|{\hat{y}}_{i}-y_{i}| \end{array}$$(4)$$\begin{array}{cc} \mathrm{PMAE} & =\frac{\mathrm{MAE}}{\frac{1}{N}\sum_{i=1}^{N}y_{i}} \end{array}$$
where $\hat{y_{i}}$ is the predicted value given the *i*-th test image, $y_{i}$ is the groundtruth value of image *i*, and MAE [32] is the mean absolute error. The values of calories are in kilocalories and the values of mass, as well as macronutrients, are in grams. The percentage of PMAE represents the percentage of mean absolute error from the average of all groundtruth values. The reason we used PMAE as our evaluation metric is that it provides a more visual representation of the accuracy of the nutritional assessment results. A lower PMAE value for each evaluation metric represents a higher accuracy of the nutritional content estimation.

## 3. Results

### 3.1. Implementation Details

We complete all experiments on a Telsa A30 GPU. We use the Adam [33] optimizer for 40 epochs using an exponentially decaying learning rate (decay rate is 0.9) scheduler. The batch size is 64, the initial learning rate is $1\times \;10^{-5}$, and the weights decayed to $1\times 10^{-5}$. We use data augmentation and regularization strategies in our training. Swin Transformer has four different model variants, namely Swin-T, Swin-S, Swin-B, and Swin-L. Considering the complexity of the model and the size of the dataset, we choose Swin-B as the backbone network and initialize our network with pre-trained weights on the ImageNet-1k dataset [34].

### 3.2. Comparison with Advanced NDDT Methods

For a fair comparison, all models are trained on the *Nutrition5k* dataset (cleaned dataset), where the train and test sets are divided in a ratio of 6:1. We use a uniform evaluation metric, PMAE, to assess the performance of the methods.

As shown in Table 1, we compare Swin-Nutrition with other state-of-the-art methods, including CNN-based methods and Transformer-based methods. We use five different network architectures: Inception V3 [35], VGG-16 [36], ResNet [30], T2T-Vit [37], and Swin Transformer [29]. We improve these networks to apply to food nutrition non-destructive detection. The descriptions of these methods are as follows:**Inception V3** is an image classification network, which consists of 11 inception modules, each of which is composed of different convolutional layers combined by parallelism. Inception V3 introduces decomposed convolution, which is significant for enriching spatial features and increasing feature diversity.**VGG-16** consists of thirteen convolutional layers and three fully connected layers, which simplify the structure of convolutional neural networks (using small convolutional kernels of 3 × 3 and maximum pooling layers of 2 × 2) and improve the performance of the network by constantly increasing the depth of the network.**ResNet** is a deeper network structure, which is composed of residual blocks and residual connections. The ResNet-101 network is mainly composed of four residual blocks, which solve the problem of gradient disappearance and gradient explosion in the deeper network.**T2T-Vit** is based on transformer structure for the image classification task, which mainly includes tokens-to-tokens block, position encoding, encoder, and MLP block. The network solves the problem that visual transformer is difficult to learn richer features by tokens-to-tokens block. The module models the local information of surrounding tokens by integrating adjacent tokens.**Swin Transformer** effectively combines the respective advantages of CNN and transformer. The method is designed for the structure of the transformer based on the hierarchical structure of CNN, and the multi-scale features for model prediction are generated by this blocked transformer module. The Swin Transformer can be widely applied to various fields of computer vision, such as image classification, image segmentation, and object detection.

Comparison with CNN-based approaches, and the comparison results are shown in Table 1. As shown in Table 1, we provide a quantitative comparison of these methods on the *Nutrition5k* dataset. The Inception V3 network predicts a caloric PMAE of 22.6%, a mass PMAE of 16.3%, and a protein PMAE of 25.1%. Compared to Inception V3, the VGG-16 improves the PMAE values for calories, mass, and protein by 0.7%, 0.3%, and 0.5%, respectively. Furthermore, we combine ResNet-101 with feature pyramid network (FPN) [38] to predict the content of five nutrients. We find that the ResNet-101+FPN network achieves better performance. The PMAE values for calories, mass, fat, carbohydrate, and protein reach 20.1%, 15.3%, 29.8%, 26.5%, and 22.9%, respectively. Compared with other networks, this difference may arise from the combination of the ResNet network with the FPN. FPN fuses the feature maps of different layers (low-level high-resolution feature maps and the high-level strong semantic feature maps) to obtain a good prediction effect.

Compared with transformer-based methods, the results of our experiments are shown in Table 1. We use the T2T-Vit network for non-destructive detection of food nutrition. As we can see, the PMAE values for calories, mass, fat, carbohydrate, and protein reach 17.0%, 13.8%, 23.8%, 22.4%, and 17.2%, respectively. In addition, we adopt Swin-B as the backbone network and pre-train it on the ImageNet-1k dataset. The model achieves good performance, where the PMAE values for calories, mass, and protein reach 16.4%, 13.3%, and 16.6%, respectively. Compared with the T2T-Vit network, the mean PMAE of the Swin Transformer decreased by 0.5%. Hence, we propose the Swin-Nutrition method based on the Swin Transformer model. The method introduces a feature fusion module, which fuses the output of the transformer module of each layer through top-down and lateral connections to obtain global and local feature information. Swin-Nutrition achieves optimal performance with PMAE values of 15.3%, 12.5%, 22.1%, 20.8%, and 15.4% for calories, mass, fat, carbohydrate, and protein, respectively.

### 3.3. Performance Analysis of the Swin-Nutrition

We perform the performance analysis of each component by removing or replacing components from the entire model of Swin-Nutrition, as shown in Table 2. The baseline is Swin-B and the uniform predicted header.

First, we explore the effectiveness of the feature fusion module. We consider that some ingredients that exist in the food dataset are relatively fine particles, which are relatively difficult to recognize by the naked eye, so we introduce the feature fusion module to obtain global and local feature information to improve the accuracy of the model prediction. Second, we analyze the effectiveness of the multi-task loss function and introduce a new loss function to solve the multi-task regression problem. Previously, the total loss value of the model is simply the sum of the loss value of the five subtasks. After introducing the multi-task loss function, we adopt adaptive loss weighting to sum the loss weights of the five subtasks. In addition, we pre-train the model using the ImageNet-1k dataset. We take the better weight values obtained at the end of pre-training as the initialized weight values for our model training, which accelerate the convergence of the model.

As shown in Table 2, the performance of the model greatly improves by adding individual components. We find that the introduction of the feature fusion module improves the overall performance of the model. Specifically, the PMAE values for calories, mass, fat, carbohydrate, and protein achieve 16.6%, 13.8%, 23.7%, 21.6%, and 16.5%, respectively. The mean value of PMAE decreased by 0.7%. The introduction of the multi-task loss function improves the performance of the model, and the mean value of PMAE decreases by 1.4%. After adding pre-training, our model achieves optimal results. The PMAE values for calories, mass, fat, carbohydrates, and protein are 15.3%, 12.5%, 22.1%, 20.8%, and 15.4%, respectively. This shows that the performance of the model is improved by 2.5% with the addition of the pre-training module. The overall performance of the method is improved by 9.5%. The mean value of PMAE achieves 17.2%.

### 3.4. Visualization and Analysis of Nutrient Prediction Results

To investigate the effectiveness of the Swin-Nutrition method, we apply the Grad-CAM (Gradient-weighted Class Activation Map) [39] operation to the last feature map of the output. In this way, the response area predicted by the model is presented more visually. The method first calculates the importance weight of feature map *A* to category *c*, then performs a weighted summation of the last layer of feature maps, and finally passes through the ReLU activation function. The method is formulated as Equations (5) and (6): (5)$$\begin{array}{cc} \alpha_{k}^{c} & =\frac{1}{Z}\sum_{i}\sum_{j}\frac{\partial y^{c}}{\partial A_{ij}^{k}} \end{array}$$(6)$$\begin{array}{cc} L_{Grad-CAM}^{C} & =ReLU\left(\sum_{k}\alpha_{k}^{c}A^{k}\right) \end{array}$$
where *c* denotes the category, *A* represents the feature maps of the convolutional output, *k* is the feature maps channel, *i* and *j* are the horizontal and vertical coordinates in the feature maps respectively, *Z* is the size of the feature maps, and $y_{c}$ represents the score predicted by the network for category *c*. The feature regions that have a positive effect on category *c* are retained using the ReLU operation.

We randomly select some images from the *Nutrition5k* dataset for visualization, as shown in Figure 4. Considering the public demand for nutritional evaluation, we visualized the predicted values of four nutrients (calories, carbohydrate, fat, and protein). Figure 4 presents us with the original images of the three dishes and the visualization results of the four metrics in each image. Furthermore, we list the various ingredients included in Dish_1560442303, and each ingredient contains nutrient values in Table 3.

For example, Dish_1560442303 contains scrambled eggs, berries, quinoa, and bacon. In Table 3, the total calorie value of Dish_1560442303 is 444.77 kcal and the calorie value of bacon reaches 205.58 kcal, which accounts for 46.22% of the total calorie value. The visualization of the calorie metric of the dish shows that the corresponding area of the calorie is in the bacon in Figure 4. This means that the model’s prediction results for calories are comparable to the groundtruth value of calories. For the prediction of fat, as can be seen in Figure 4, the fat response areas are located on scrambled eggs and bacon. We find that the fat content of scrambled eggs and bacon accounts for 95.5% of the total fat content, in Table 3. Carbohydrate is mainly present in berries and quinoa, which account for 94.2% of the total carbohydrate content. We observe that the response region of the carbohydrate is also located in berries and quinoa. This indicates that the model’s prediction results for carbohydrate are correct. In the same way, we can observe other images on the *Nutrition5k* dataset. We find that fat loss values are relatively high compared to the other four metrics, in Table 1. After studying the relevant data, most dishes have edible oils such as olive oil and vegetable oil, and the main component of these edible oils is fat. Edible oils adhere to the surface of ingredients and are difficult for humans to observe with the naked eye. These factors contribute to the poor predictive performance of fat. Therefore, the prediction of fat remains challenges. The visualization results show that the response area of each metric is accurately predicted in the image. This also confirms the effectiveness of our method. Our Swin-Nutrition method can evaluate the calorie, mass, and three macronutrient values in food, and outperforms other methods.

## 4. Discussion

Swin-Nutrition was confirmed experimentally to be a competitive model for the nutritional evaluation of the various categories of foods taken in the daily diet. We utilized this model to evaluate the content of calories, carbohydrates, fats, and proteins in the food. In the evaluation results, we found that the loss values of calories, mass, carbohydrates, fats, and proteins were only 15.3%, 12.5%, 20.8%, 22.1%, and 15.4%, respectively. In previous studies, Thames et al. [21] have also confirmed the feasibility and accuracy of computer vision-based nutritional evaluation. Similarly, the visualization results of the Swin-Nutrition method also showed that the model responded correctly to the corresponding regions for each metric. Compared with the mainstream methods, Swin-Nutrition also achieved the best performance for prediction. Swin-Nutrition utilized a powerful feature extraction and feature fusion module to enhance feature representation, which addressed the difficulty of detecting fine ingredients for food nutrition assessment.

Although deep learning-based techniques for non-destructive detection of food nutrients are emerging, there are still many challenges against their widespread applications. First, physical (sensor) and chemical food detection techniques have become more mature but cannot be widely applied in public daily life. With the wide application of artificial intelligence technology, deep learning-based food NDDT is a challenging task. Secondly, the current food NDDT technology [40,41] simply predicts and evaluates various food nutrients, and the accuracy needs to be considered. In addition, it is still far from being widely used in the daily life of the public. In addition, computer vision-based methods expose their limitations when food is stacked on top of one another. Therefore, future research needs to explore cross-modal and cross-domain nutritional evaluation methods. Finally, there are fewer available datasets for food nutrition evaluation. Data power the development of artificial intelligence, and the development of all emerging technologies is inseparable from the support of data. The process of collecting, constructing, and labeling nutrition data are complex and requires many human resources. Despite these challenges, the great potential of deep learning-based methods for food nutrition non-destructive detection remains to be explored.

In future work, we will create a food dataset with richer nutrition labels to contribute to food nutrition research. In addition, we will continue to explore more accurate methods for non-destructive detection of food nutrition to meet people’s needs for diet monitoring and ensure a balanced diet for the public.

## 5. Conclusions

We propose the Swin-Nutrition method for fast and non-destructive detection of food nutrition content. Swin-Nutrition integrates the Swin Transformer with a pyramid-structured feature fusion module to enhance the accuracy of nutrition content detection. We design a pyramid-structured feature fusion module to fuse multi-scale feature maps and capture powerful feature representations. Swin-Nutrition aims to explore an efficient and fast method for nutrition non-destructive detection, which extracts feature information through the Swin-B backbone network and obtains rich local and global feature information through the FFM module to improve the accuracy of non-destructive detection. We visualize the predicted results for each nutrient and confirm the efficiency of our method on the *Nutrition5k* dataset. In the future, we hope that automated food nutrition non-destructive detection methods will be widely used in the daily diet of the public.

## Figures and Tables

**Figure 1 foods-11-03429-f001:**
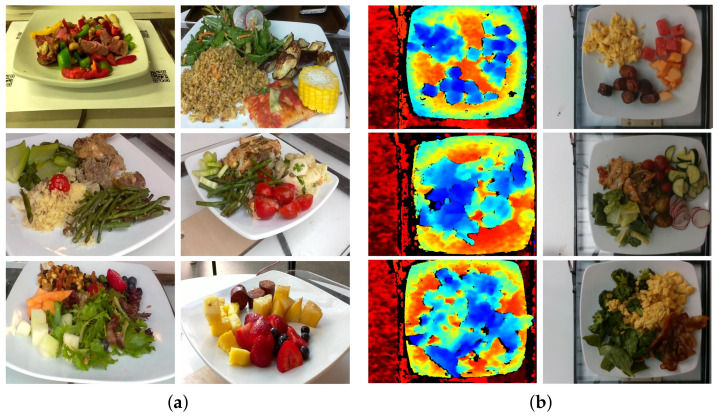
(**a**) RGB images from *Nutrition5k* [21]; (**b**) depth images from *Nutrition5k* [21].

**Figure 2 foods-11-03429-f002:**
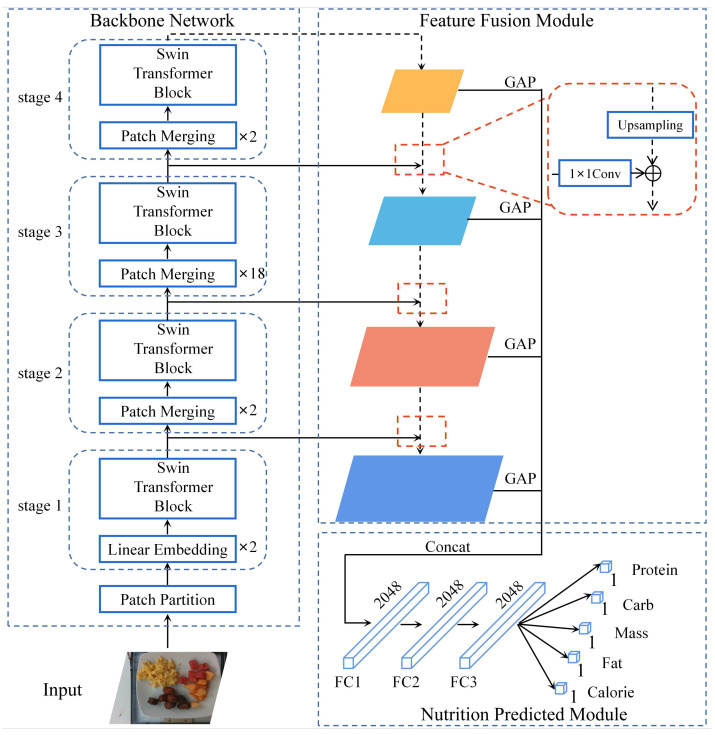
The overall architecture of Swin-Nutrition. The network mainly consists of backbone network, feature fusion module (FFM), and nutrition prediction module. GAP denotes global average pooling and FC means fully connected layer.

**Figure 3 foods-11-03429-f003:**
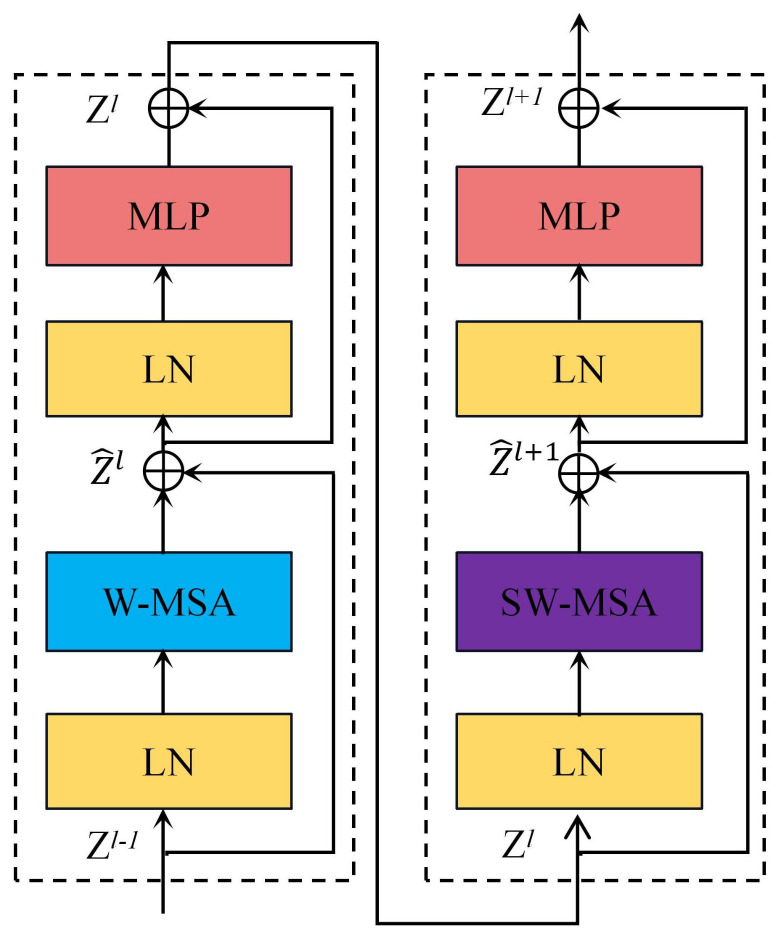
Detailed structure of the Swin Transformer block. W-MSA denotes window-based self-attention, and SW-MSA represents shifted window-based self-attention. ${\hat{Z}}^{i}$ and $Z^{i}$ denote the output feature map of the *i*-th block after (S)W-MSA and MLP, respectively.

**Figure 4 foods-11-03429-f004:**
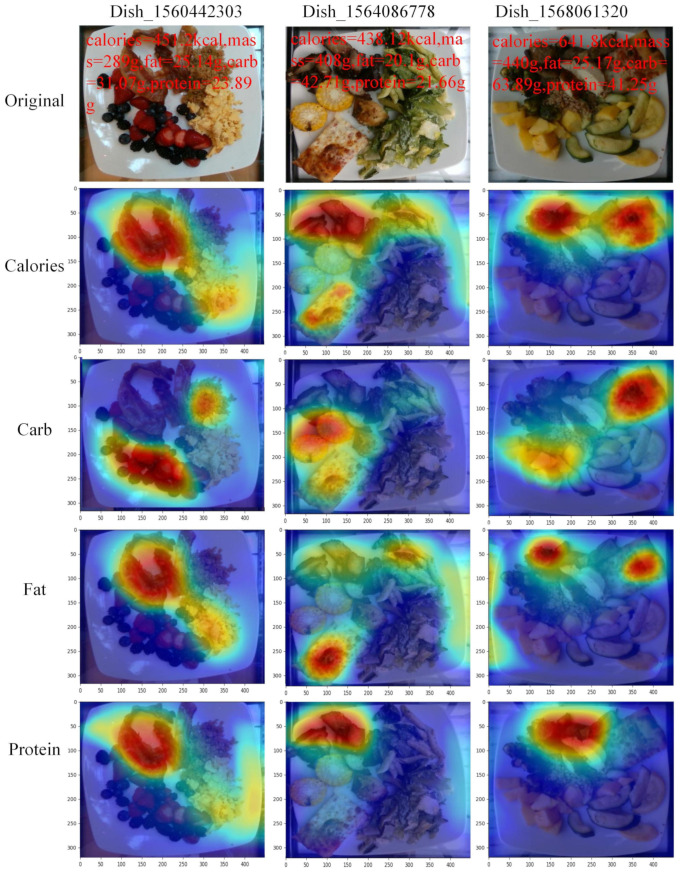
Visualization results. We randomly select three images and visualize them with the Grad-CAM method.

**Table 1 foods-11-03429-t001:** The comparison between Swin-Nutrition and other methods. The best results are shown in bold.

Method Types	Methods	Backbone	Pretain	Calories PMAE (%)	Mass PMAE (%)	Fat PMAE (%)	Carb. PMAE (%)	Protein PMAE (%)	Mean PMAE (%)
CNN-based methods	Inception V3 [35]	Inception V3	N	22.6	16.3	32.5	27.5	25.1	24.8
	VGG-16 [36]	VGG-16	N	21.9	16.0	32.1	28.0	24.6	24.5
	ResNet-101+FPN [30]	ResNet-101	N	20.1	15.3	29.8	26.5	22.9	22.9
transformer- based methods	T2T-ViT [37]	T2T-ViT	N	17.0	13.8	23.8	22.4	17.2	18.8
	Swin Transformer [29]	Swin-B	Y	16.4	13.3	23.6	21.6	16.6	18.3
	Swin-Nutrition	Swin-B	Y	**15.3**	**12.5**	**22.1**	**20.8**	**15.4**	**17.2**

**Table 2 foods-11-03429-t002:** Performance comparison results for each module of our method on the *Nutrition5k* dataset. The best results are shown in boldface.

Method			*Nutrition5k*					
**FFM**	**Multi-Task** **Loss**	**Pretrain-ing**	**Calories** **PMAE (%)**	**Mass** **PMAE (%)**	**Fat** **PMAE (%)**	**Carb.** **PMAE (%)**	**Protein** **PMAE (%)**	**Mean** **PMAE (%)**
			17.1	14.6	24.3	22.3	17.2	19.1
✓			16.6	13.8	23.7	21.6	16.5	18.4
✓	✓		15.8	12.9	23.0	21.1	15.9	17.7
✓	✓	✓	**15.3**	**12.5**	**22.1**	**20.8**	**15.4**	**17.2**

**Table 3 foods-11-03429-t003:** Nutrition content annotation information of Dish_1560442303.

Ingredient	Cal (kcal)	Fat (g)	Carb. (g)	Protein (g)
scrambled eggs	111	8.25	1.2	7.5
berries	83.79	0.44	20.58	1.03
quinoa	44.4	0.70	7.77	1.63
bacon	205.58	15.96	0.53	14.06
Sum	444.77	25.35	30.08	24.22

## Data Availability

Not applicable.

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
