# Peer review of "Rapid Non-Destructive Analysis of Food Nutrient Content Using Swin-Nutrition"

_foods, 2022, doi:10.3390/foods11213429_

Round 1
Reviewer 1 Report
The manuscript is well written (consider that I am not a native speaker), clearly structured and quite well organized. Without doubt, the information gathered with the present study will attract a great deal of food and nutrition field, as well as the academic interest. A thorough description of the actual scenario of the scientific research already performed in this frame is provided by the authors.
Moreover, I think the present manuscript fit with the scope of the Issue and the Review. I came across some parts that need revision (in form of corrections/integrations). Details of them are listed below.
General comments
1. What about the authors decision of directly considering a so complex deep learning-based method instead of first investigating a simpler and, maybe, interpretable approach, based on linear chemometrics techniques, like for example exploratory PCA followed by classification and/or regression with PLS-DA or PLS algorithms? Are there some conditions that lead to interpret the problem under investigation as a “non-linear” case? How do authors perceived/established it?
2. I can imagine that authors have faced with the most common problem that usually we have to deal with when processing and analysing any kind of images: the background removal. How has this preliminary step been tackled?
3. It is not so clearly explained how the analytical reference values (called as "ground truth values" from the authors) of the 5 food macronutrients/features (i.e., fat, carbohydrates, proteins, calories and mass)? were obtained? Can they also be found on the site where the "Nutrition 5k dataset" come from? In particular, how the parameter "mass" could be predicted in this work? Where do mass reference values come from?
4. Do you have an idea of which wavelengths, or wavelength regions, better performed to achieve the promising results presented in Tables 1 and 2? According to my knowledge and expertise in image and spectroscopy analysis, I can say that physical-chemical information on food matrices can be gathered mainly by the infrared part of the spectrum, rather than by the RGB wavelengths. But, in any case, I think that some considerations could be done by authors about this point.
Targeted comments
Introduction, Line 30: authors say “…food nutrition detection”, but I think that “…food nutrients detection” sounds a little bit better. This expression is used also in following parts of the manuscript. Please, have a check.
Materials and Methods, Lines 98-101: authors talk about the camera sensors, but it would be better to specify the wavelengths, or the wavelengths range, covered by the infrared and colour sensors.
Section 2.2.2, Line 138: authors say “…changed by C”, but what they exactly mean?
Table 2: which method was it used to achieve the results reported in the first row? This information should at least be embedded in the table's caption.
Section 3.4, Line 315: it is not clear what authors exactly mean here saying "…the four metrics". Are they referring to the evaluation metrics described in Section 2.3 or to different ones? Did other evaluation metrics apart MAE and PMAE were considered in this study?
Section 3.4, Line 330: why do authors talk about "loss values" instead of simply "values"? Are the loss values directly related to the PMAE values? Please, try to better clarify this point.
Section 3.4, Lines 332-334. Ok, it is true. Thus, it can be assumed that RGB spectral regions are not so effective to fix the problem at hand. But what about the chemical-physical features that could be gathered by the two infrared sensors (tying in with the general comment number 4)?
Section 3.4, Lines 334-335: I suggest changing the sentence in "Therefore, the prediction of fat remains a challenge or "Therefore, the prediction of fat remains challenging".
Figure 4: since each subfigure (apart the up three) represent as a kind of "intensity map" obtained as a prediction result of each considered macronutrient, I suggest adding, near each of them, a suitable colormap with the intensity values, in order to get the reader a clearer and immediate quantitative interpretation of the results.
Moreover, what about the prediction of the 5th food feature considered, the mass? In this case, is it not possible to make any kind of graphical representation of the modelling results?
Discussion, Line 342: authors did not mention the prediction of the "mass" in the Discussion section. And anything is no longer said about this investigated parameter. Why?
Discussion, Lines 352-373: this part do not seem to be a discussion directly related to the results achieved in the present study and/or to a direct comparison of the results with those achieved by other similar research studies. It covers quite general aspects, that fit more as Conclusion than Discussion section. Moreover, lines from 370 to 373 could be a valid Perspective. Thus, I suggest authors to revise and rearranging this part, maybe by merging the consideration herein included with those reported in the Conclusions section.
Author Response
Dear reviewer,
We highly appreciate your valuable suggestions and comments in improving our manuscript. We have given careful consideration to all the comments and revised our manuscript accordingly.
In the followings, we present our detailed response to each comment by listing the original comments using black texts followed by our response shown in red font.
Best regards,
Sujuan Hou

Reviewer 2 Report
I reviewed the manuscript entitled, Rapid Non-destructive Analysis of Food Nutrient Content Using Swin-Nutrition. The study deals with the determination of food nutrient composition using Shifted Windows (\textbf{S}hifted \textbf{win}dows) Transformer based method. The research is novel and contributes to the field. The approaches authors used may motivate researchers/the general public to use on a daily basis. However, authors also provided the limitation of the study. Based on these observations, I would like to recommend the manuscript after considering the minor suggestions.
Line 26: Introduce what is NDDT for the first time in Introduction
Lines 88-94: should move to conclusions
Lines 78-84: it seems like background info Swin nutrition. Please move to before research objectives
Lines 84-87: it's not an appropriate place. Please move it
Figure 1. Please provide copyright info
Lines 113-117: it is redundant; please remove it
Line 210: please give the space between “Adam” and “37”
References are not according to the journal format
Other results and discussion sections are appropriate; however, discussion must be improved with supportive literature.
Author Response

(The authors gave the same response as above.)
